# Rubik's Cube: High-Order Channel Interactions with a Hierarchical Receptive Field

**Naishan Zheng**[*]   **Man Zhou***   **Chong Zhou**   **Chen Change Loy**
S-Lab, Nanyang Technological University
naishan.zheng,man.zhou,chong033,ccloy@ntu.edu.sg

## Abstract

Image restoration techniques, spanning from the convolution to the transformer paradigm, have demonstrated robust spatial representation capabilities to deliver high-quality performance. Yet, many of these methods, such as convolution and the Feed Forward Network (FFN) structure of transformers, primarily leverage the basic first-order channel interactions and have not maximized the potential benefits of higher-order modeling. To address this limitation, our research dives into understanding relationships within the channel dimension and introduces a simple yet efficient, high-order channel-wise operator tailored for image restoration. Instead of merely mimicking high-order spatial interaction, our approach offers several added benefits: *Efficiency*: It adheres to the zero-FLOP and zero-parameter principle, using a spatial-shifting mechanism across channel-wise groups. *Simplicity*: It turns the favorable channel interaction and aggregation capabilities into element-wise multiplications and convolution units with $1 \times 1$ kernel. Our new formulation expands the first-order channel-wise interactions seen in previous works to arbitrary high orders, generating a hierarchical receptive field akin to a Rubik's cube through the combined action of shifting and interactions. Furthermore, our proposed Rubik's cube convolution is a flexible operator that can be incorporated into existing image restoration networks, serving as a drop-in replacement for the standard convolution unit with fewer parameters overhead. We conducted experiments across various low-level vision tasks, including image denoising, low-light image enhancement, guided image super-resolution, and image de-blurring. The results consistently demonstrate that our Rubik's cube operator enhances performance across all tasks. Code is publicly available at https://github.com/zheng980629/RubikCube.

## 1 Introduction

Image restoration aims at restoring a high-quality image from its degraded counterpart. This task is inherently ill-posed, given that a multitude of valid solutions could potentially exist for a single degraded image. The convolutional neural network (CNN), with its remarkable learning capabilities, has been a driving force behind the surge of learning-driven methodologies for image restoration, including the residual block [1, 2], dense connection [3, 4], multi-stage strategy [5, 6], and dropout [7].

**High-order spatial modeling.** As shown by recent studies [8, 9, 10, 11, 12], vision transformers have overtaken convolutional neural networks as the leading method for image restoration. The key factor driving this shift is the powerful mechanism for modeling spatial interactions inherent in vision transformers. Unlike the local first-order spatial modeling found in the convolution family [13, 14, 15], vision transformers [16] employ a dot-product self-attention paradigm. This paradigm performs matrix multiplication among queries, keys, and values, thereby enhancing the model's capacity by

---

[*]These authors contributed equally to the work.

37th Conference on Neural Information Processing Systems (NeurIPS 2023).

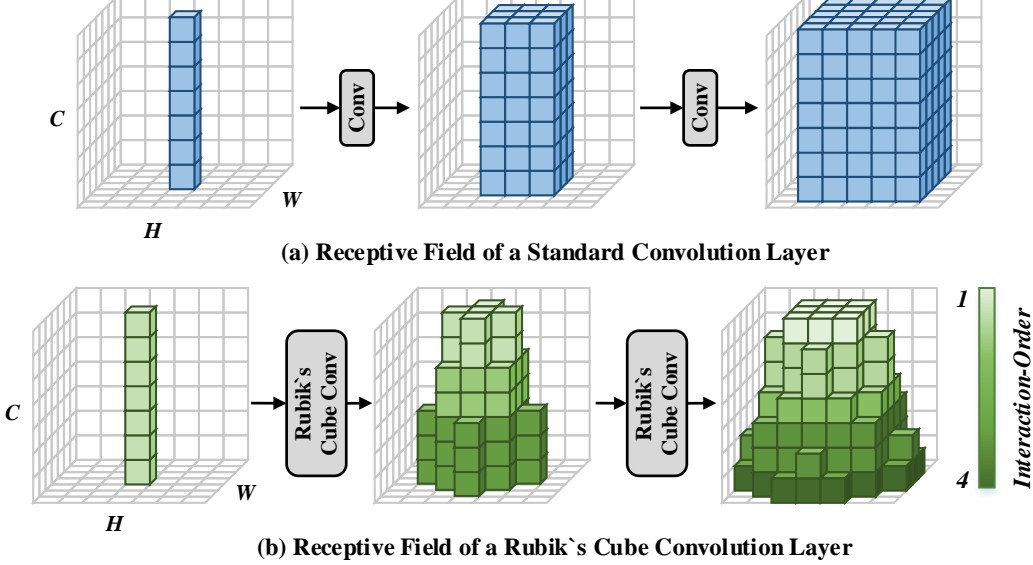

**(a) Receptive Field of a Standard Convolution Layer**

**(b) Receptive Field of a Rubik`s Cube Convolution Layer**

Figure 1: **Comparison between the vanilla convolution and the proposed Rubik's cube convolution operator.** Different from the previous first-order linear weighting in the channel dimension, the proposed operator formulates high-order channel interactions and produces a Rubik's cube-like hierarchical receptive field (the receptive field expands as the order of channel interaction increases).

two successive global spatial interactions. Rao et al. [17] investigate the reasons behind the superior performance of vision transformers compared to convolutional networks. The study concludes that the credit predominantly goes to the high-order spatial interaction modeling of vision transformers.

**Motivation.** While significant strides have been made in exploring the spatial dimension, the interaction inherent in the alternative channel dimension remains largely untapped. Specifically, most existing image restoration methods, such as the convolution unit and the transformer's FFN layer (which can be considered as a $1 \times 1$ convolution), only benefit from the first-order channel interaction through linear channel-wise weighting, leaving ample opportunity for exploring high-order modeling. This observation motivates us to explore the modeling of channel-dimension relationships.

**Solution.** The primary goal of this paper is to examine the potential of channel interactions and propose an alternative operator for high-order channel-wise modeling in the context of image restoration. To achieve rich and meaningful channel interactions, we introduce a high-order channel-wise operator that is simple, effective and efficient. Specifically, instead of merely imitating high-order spatial interaction, our operator is uniquely formulated on the principles of a zero-FLOP and zero-parameter spatial-shifting mechanism, which is applied over channel-wise groups. We achieve high-order channel interactions by performing element-wise multiplications among these groups. Our newly proposed operator extends beyond the first-order channel-wise interactions found in prior works, raising them to arbitrary orders without introducing significant computational overhead. Through the combination of shifting and channel interactions, our operator constructs a hierarchical, Rubik's cube-like receptive field (see Figure 1). As such, we have dubbed our operator the Rubik's cube convolution.

The key contribution of this paper is a novel high-order operator featuring a Rubik's cube-like hierarchical receptive field. The Rubik's cube convolution we propose can be seamlessly integrated into existing image restoration networks, serving as a drop-in replacement for the standard convolution unit, but with a reduced parameter count. Comprehensive experiments across various low-level tasks, including image denoising, low-light image enhancement, guided image super-resolution, and image de-blurring, demonstrate the effectiveness of our operator.

## 2 Related Work

**Image Restoration.** Image restoration is tasked with the recovery of a latent clean image from observations corrupted by degrading factors such as noise [18, 19, 20], blur [21, 22], or inadequate lighting conditions [23, 24, 25, 26, 27]. It is a fundamentally ill-posed problem, as an infinite number of feasible solutions can be conceived for a single degraded image. Traditional image restoration techniques address this issue by formulating the task as an optimization problem, assuming specific priors of the latent high-quality images to regularize the solution space. For instance, the low-rank prior [28] and total variation regularization [29] have been devised for image deblurring, while histogram distribution prior [30] and bright channel prior [31] have been developed for low-light image enhancement. Despite their effectiveness, these hand-crafted priors are challenging to design and often exhibit limited representational capabilities in complex scenarios, restricting their practical applicability. The advent of deep learning has sparked a surge in learning-based algorithms within the image restoration community. Early works in this domain, such as SRCNN [32] for super-resolution and DnCNN [33] for image denoising, leveraged stacked convolution layers to capture effective feature representations. Subsequent research introduced various powerful model designs, such as residual blocks [1, 2], dense connections [3], multi-stage strategies [5], and dropout [7], to image restoration tasks. Furthermore, researchers have begun to probe into spatial and channel interaction mechanisms [34, 35, 9], aiming to excavate more meaningful degradation and content cues within the feature space of image restoration networks.

**Channel-Dimension Modeling.** While the self-attention mechanism within the Transformer [16] has shown the importance of modeling spatial interaction for vision tasks, fewer studies have explored the channel relationship, especially in image restoration tasks. Zhang *et al.*[36] integrated channel-wise attention to enhance performance in super-resolution. Chen *et al.*[6] applied instance normalization to half of the intermediate features and kept the remaining content information untouched to improve model capacity. Li *et al.* [37] performed hybrid-distorted image restoration using a feature-level divide-and-conquer strategy, disentangling the feature representation of different distortions into distinct channels. Existing approaches typically benefit explicitly from first-order channel-wise interaction or channel disentanglement, neglecting its potential for high-order interaction.

## 3 Rubik's Cube

### 3.1 Revisit Channel Interaction

The global spatial interaction in the self-attention family [8, 9, 11, 12] has validated its effectiveness in the image restoration community. Hornet [17] further demonstrates that the key ingredient behind its success is the high-order spatial interaction modeling.

Deepening into the channel dimension, we first revisit channel interaction modeling within the CNN and transformer families. The standard convolution layer and the inherent FFN layer in transformers only apply linear channel-wise weighting to the input feature. The Squeeze-and-Excitation (SE) block [38, 39, 40, 41] accomplishes first-order channel interaction by adaptively recalibrating channel-wise feature responses using the squeezed input feature. Restormer [10], with the self-attention mechanism, carries out two consecutive first-order interactions in the channel dimension by conducting matrix multiplication among queries, keys, and values. These operations only explore the first-/second-order channel interaction implicitly.

### 3.2 Rubik's Cube Convolution

In order to devise an operator that effectively and efficiently facilitates high-order interaction among channels, we turn to the principles of spatial shifting, a mechanism that operates with zero FLOPs and does not require any parameters. We apply this spatial shifting to each group with channel dimension, thereby enabling high-order channel interaction. The details of the shift operations, high-order channel interaction, and the associated receptive field are discussed in the following paragraphs.

**Shift Operation.** As shown in Figure 2, given an input feature map $\mathbf{X} \in \mathbb{R}^{H \times W \times C}$, we evenly divide $\mathbf{X}$ into five parts by the channel dimension, where the first is kept unchanged and the remaining four ones are shifted in a distinct spatial direction: left, right, top, and down. Subsequent to the shifting operation, we discard out-of-focus pixels and any vacant pixels are filled with zeros.

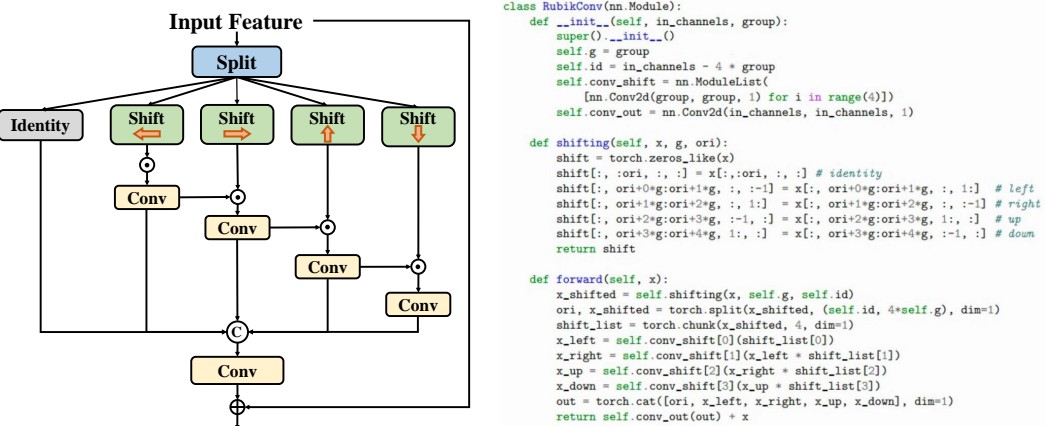

```python
class RubikConv(nn.Module):
    def __init__(self, in_channels, group):
        super().__init__()
        self.g = group
        self.id = in_channels - 4 * group
        self.conv_shift = nn.ModuleList(
            [nn.Conv2d(group, group, 1) for i in range(4)])
        self.conv_out = nn.Conv2d(in_channels, in_channels, 1)

    def shifting(self, x, g, ori):
        shift = torch.zeros_like(x)
        shift[:, :ori, :, :] = x[:,:ori, :, :] # identity
        shift[:, ori+0*g:ori+1*g, :, :-1] = x[:, ori+0*g:ori+1*g, :, 1:] # left
        shift[:, ori+1*g:ori+2*g, :, 1:]  = x[:, ori+1*g:ori+2*g, :, :-1] # right
        shift[:, ori+2*g:ori+3*g, :-1, :] = x[:, ori+2*g:ori+3*g, 1:, :]  # up
        shift[:, ori+3*g:ori+4*g, 1:, :]  = x[:, ori+3*g:ori+4*g, :-1, :] # down
        return shift

    def forward(self, x):
        x_shifted = self.shifting(x, self.g, self.id)
        ori, x_shifted = torch.split(x_shifted, (self.id, 4*self.g), dim=1)
        shift_list = torch.chunk(x_shifted, 4, dim=1)
        x_left = self.conv_shift[0](shift_list[0])
        x_right = self.conv_shift[1](x_left * shift_list[1])
        x_up = self.conv_shift[2](x_right * shift_list[2])
        x_down = self.conv_shift[3](x_up * shift_list[3])
        out = torch.cat([ori, x_left, x_right, x_up, x_down], dim=1)
        return self.conv_out(out) + x
```

Figure 2: **Overview of the Rubik's cube convolution layer.** It stands on the principle of zero-FLOP and zero-parameter spatial-shifting mechanism over channel-wise groups. It achieves high-order channel interactions with element-wise multiplications and convolution layers with $1 \times 1$ kernel, rendering it *simple* and *efficient*. We also provide the corresponding PyTorch-style code on the right.

The shifted feature $\hat{\mathbf{X}}$ can be written as:

$$\hat{\mathbf{X}}[0:\mathrm{H}, 0:\mathrm{W}, 0:\mathbf{C}_{\mathrm{id}}] \leftarrow \mathbf{X}[0:\mathrm{H}, 0:\mathrm{W}, 0:\mathbf{C}_{\mathrm{id}}],$$

$$\hat{\mathbf{X}}[0:\mathrm{H}, 1:\mathrm{W}, \mathbf{C}_{\mathrm{id}}:\mathbf{C}_{\mathrm{id}}+\mathbf{C}_{\mathrm{g}}] \leftarrow \mathbf{X}[0:\mathrm{H}, 0:\mathrm{W}-1, \mathbf{C}_{\mathrm{id}}:\mathbf{C}_{\mathrm{id}}+\mathbf{C}_{\mathrm{g}}],$$

$$\hat{\mathbf{X}}[0:\mathrm{H}, 0:\mathrm{W}-1, \mathbf{C}_{\mathrm{id}}+\mathbf{C}_{\mathrm{g}}:\mathbf{C}_{\mathrm{id}}+2\mathbf{C}_{\mathrm{g}}] \leftarrow \mathbf{X}[0:\mathrm{H}, 1:\mathrm{W}, \mathbf{C}_{\mathrm{id}}+\mathbf{C}_{\mathrm{g}}:\mathbf{C}_{\mathrm{id}}+2\mathbf{C}_{\mathrm{g}}], \tag{1}$$

$$\hat{\mathbf{X}}[0:\mathrm{H}-1, 0:\mathrm{W}, \mathbf{C}_{\mathrm{id}}+2\mathbf{C}_{\mathrm{g}}:\mathbf{C}_{\mathrm{id}}+3\mathbf{C}_{\mathrm{g}}] \leftarrow \mathbf{X}[1:\mathrm{H}, 0:\mathrm{W}, \mathbf{C}_{\mathrm{id}}+2\mathbf{C}_{\mathrm{g}}:\mathbf{C}_{\mathrm{id}}+3\mathbf{C}_{\mathrm{g}}],$$

$$\hat{\mathbf{X}}[1:\mathrm{H}, 0:\mathrm{W}, \mathbf{C}_{\mathrm{id}}+3\mathbf{C}_{\mathrm{g}}:\mathbf{C}_{\mathrm{id}}+4\mathbf{C}_{\mathrm{g}}] \leftarrow \mathbf{X}[0:\mathrm{H}-1, 0:\mathrm{W}, \mathbf{C}_{\mathrm{id}}+3\mathbf{C}_{\mathrm{g}}:\mathbf{C}_{\mathrm{id}}+4\mathbf{C}_{\mathrm{g}}],$$

where $\mathbf{C}_{\mathrm{id}}$ is the number of channels of the unchanged identity part, $\mathbf{C}_{\mathrm{g}}$ is the number of channels of a shifted group, and $\mathbf{C}_{\mathrm{id}}+4*\mathbf{C}_{\mathrm{g}}=\mathbf{C}$. In this work, the default shift of pixel is set to 1, and its robustness is validated in Sec. 4.4. Next, the shifted feature $\hat{\mathbf{X}}$ is split into $\hat{\mathbf{X}}_{\mathrm{ori}} \in \mathbb{R}^{\mathrm{H} \times \mathrm{W} \times \mathbf{C}_{\mathrm{id}}}$ and $\{\hat{\mathbf{X}}_{\mathrm{c}1}, \hat{\mathbf{X}}_{\mathrm{c}2}, \hat{\mathbf{X}}_{\mathrm{c}3}, \hat{\mathbf{X}}_{\mathrm{c}4}\} \in \mathbb{R}^{\mathrm{H} \times \mathrm{W} \times \mathbf{C}_{\mathrm{g}}}$ along the channel dimension. The zero-FLOP and zero-parameter shifting operation ensure the *efficiency* of the proposed Rubik's cube convolution.

**High-order Channel Interaction.** After the shifting and grouping operations, we proceed to model the channel-wise relationship as depicted in Figure 3. More precisely, we use a convolution layer with a $1 \times 1$ kernel to consolidate information within a shifted group. Subsequently, we facilitate a first-order channel-wise interaction by performing an element-wise multiplication with the succeeding shifted group. The first-order channel interaction can be written as follows:

$$\widetilde{\mathbf{X}}_{\mathrm{c}1} = \mathrm{Conv}_{1 \times 1}(\hat{\mathbf{X}}_{\mathrm{c}1}),$$
$$\widetilde{\mathbf{X}}_{\mathrm{c}2} = \mathrm{Conv}_{1 \times 1}(\widetilde{\mathbf{X}}_{\mathrm{c}1} \odot \hat{\mathbf{X}}_{\mathrm{c}2}), \tag{2}$$

where $\odot$ indicates the element-wise multiplication. Then, the high-order channel interaction can be successively achieved as:

$$\widetilde{\mathbf{X}}_{\mathrm{c}3} = \mathrm{Conv}_{1 \times 1}(\widetilde{\mathbf{X}}_{\mathrm{c}2} \odot \hat{\mathbf{X}}_{\mathrm{c}3}),$$
$$\widetilde{\mathbf{X}}_{\mathrm{c}4} = \mathrm{Conv}_{1 \times 1}(\widetilde{\mathbf{X}}_{\mathrm{c}3} \odot \hat{\mathbf{X}}_{\mathrm{c}4}). \tag{3}$$

Finally, we aggregate the original unchanged feature and the advanced feature after the high-order channel interaction to obtain the result of the Rubik's cube convolution:

$$\mathbf{X}_{\mathrm{out}} = \mathrm{Conv}_{1 \times 1}(\mathrm{Concat}[\hat{\mathbf{X}}_{\mathrm{ori}}, \widetilde{\mathbf{X}}_{\mathrm{c}1}, \widetilde{\mathbf{X}}_{\mathrm{c}2}, \widetilde{\mathbf{X}}_{\mathrm{c}3}, \widetilde{\mathbf{X}}_{\mathrm{c}4}]) + \mathbf{X}. \tag{4}$$

The details of the designed operator and its pseudo-code are summarized in Figure 2 The proposed Rubik's cube convolution achieves high-order channel interaction and information aggregation by element-wise multiplication and convolution layers with $1 \times 1$ kernel, which is *simple* to implement.

**Rubik's Cube-like Hierarchical Receptive Field.** Examining from the channel dimension, our proposed Rubik's cube convolution segregates the input feature into five non-overlapping groups, thereby establishing element-wise interaction and information aggregation among them. This process encourages the attainment of high-order channel-wise interactions in a simple and effective manner.

Shifting our focus to the spatial dimension, as depicted in Figure 3, we observe that the coupling of the shifting operation with channel interaction progressively expands the receptive field as the channel-wise interaction order increases. This results in a stratified receptive field along the channel dimension, reminiscent of a Rubik's cube. As illustrated in Figure 4, the effective receptive field [42] of each group successively broadens post-shifting in each direction. Moreover, it is worth noting that the effective receptive field undergoes further expansion with an increase in the pixels shifted. This mechanism is further dissected in Sec. 4.3 and Figure 6.

### 3.3 Parameter Analysis

We conduct the complexity analysis between the standard convolution layer with a $3 \times 3$ kernel and our proposed operator. Assuming the number of input channels and output channels is identical, i.e., $C \to C$ and the ratio of the shifted channel is equal to $1/2$ (default setting in all experiments and its robustness is verified in Sec. 4.4) as an example. The required number of trainable parameters is:

$$\text{Conv}_{\text{para}} = 3 \times 3 \times C \times C = 9C^2 \tag{5}$$

The proposed Rubik's cube convolution is implemented by convolution layers with $1 \times 1$ kernel over the shifted groups, where the trainable parameters are:

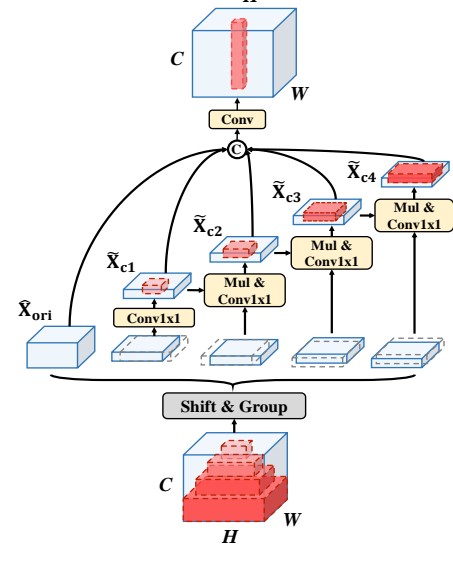

Figure 3: **Illustration of the Rubik's cube-like hierarchically receptive field.** Through the combined action of shifting and interactions, the effective receptive field [42] of each group gradually expands with the order of channel-wise interaction increases. The red-shaded region indicates the receptive field of the corresponding group.

$$\text{RubikConv}_{\text{para}} = 4 \times (1 \times 1 \times \frac{C}{8} \times \frac{C}{8}) + 1 \times 1 \times C \times C = \frac{17}{16}C^2 \tag{6}$$

Therefore, the parameter ratio is

$$\frac{\text{RubikConv}_{\text{para}}}{\text{Conv}_{\text{para}}} = \frac{17}{144}. \tag{7}$$

## 4 Experiments

We conduct extensive experiments on multiple image restoration and enhancement tasks to demonstrate the effectiveness of our proposed Rubik's cube convolution. We provide more experimental results on the recognition tasks in the supplementary material.

### 4.1 Experimental Settings

**Image Enhancement.** We conducted experiments on two widely used low-light image enhancement datasets: LOL [43] and Huawei [44]. The LOL dataset consists of 500 paired low-/normal images,

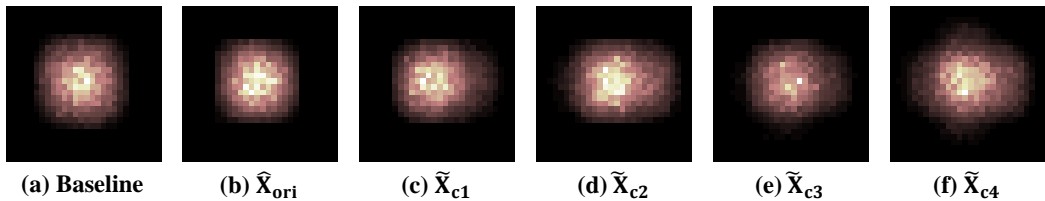

(a) Baseline     (b) $\hat{\mathbf{X}}_{\text{ori}}$     (c) $\widetilde{\mathbf{X}}_{c1}$     (d) $\widetilde{\mathbf{X}}_{c2}$     (e) $\widetilde{\mathbf{X}}_{c3}$     (f) $\widetilde{\mathbf{X}}_{c4}$

Figure 4: **Visualization of the effective receptive field [42].** (a) The baseline indicates the effective receptive field of the default architecture, and the last five describe the effective receptive field of the (b) identity group, $\hat{\mathbf{X}}_{\text{ori}}$, (c) left-shifting group, $\widetilde{\mathbf{X}}_{c1}$, (d) right-shifting group, $\widetilde{\mathbf{X}}_{c2}$, (e) up-shifting group, $\widetilde{\mathbf{X}}_{c3}$, (f) and down-shifting group, $\widetilde{\mathbf{X}}_{c4}$ after interactions in the Rubik's cube convolution.

Table 1: Quantitative comparison of low-light image enhancement on the LOL [43] and Huawei [44] datasets. A → B means generalization whose source dataset is A and the target dataset is B.

| Model | Config | LOL → | | → Huawei | | Huawei → | | → LOL | | #Params |
|---|---|---|---|---|---|---|---|---|---|---|
| | | PSNR | SSIM | PSNR | SSIM | PSNR | SSIM | PSNR | SSIM | |
| DRBN | Original | 19.7931 | 0.8361 | 17.7929 | 0.6247 | 20.1549 | 0.6851 | 18.0856 | 0.7543 | 0.55M |
| | Conv1x1 | 19.8648 | 0.8340 | 16.2748 | 0.6118 | 20.0189 | 0.6854 | 17.9643 | 0.7518 | 0.55M |
| | RubikConv | **20.3769** | **0.8400** | **17.9811** | **0.6337** | **20.2363** | **0.6876** | **18.2066** | **0.7543** | 0.55M |
| SID | Original | 20.1062 | 0.7895 | 16.5874 | 0.5925 | 20.1742 | 0.6659 | 18.5468 | 0.7441 | 7.76M |
| | Conv1x1 | 19.9709 | 0.7753 | 16.3357 | 0.5780 | 20.1742 | 0.6659 | 18.5468 | 0.7441 | 7.71M |
| | RubikConv | **20.4972** | **0.7979** | **16.7867** | **0.5941** | **20.2044** | **0.6665** | **18.6600** | **0.7476** | 7.69M |

Table 2: Quantitative comparison of image de-bluring. The model is only trained on the GoPro [45] training set and directly tested on the GoPro [45] testing set, HIDE [46], and RealBlur [47] datasets.

| Model | Metric | DeepDeblur | | | MPRNet | | | Restormer | | |
|---|---|---|---|---|---|---|---|---|---|---|
| | | Original | Conv1x1 | RubikConv | Original | Conv1x1 | RubikConv | Original | Conv1x1 | RubikConv |
| GoPro | PSNR | 28.9423 | 28.8247 | **29.1919** | 32.6546 | 32.5638 | **32.7197** | 32.9117 | 32.7920 | **32.9305** |
| | SSIM | 0.8716 | 0.8708 | **0.8761** | 0.9576 | 0.9566 | **0.9579** | 0.9603 | 0.9590 | **0.9608** |
| HIDE | PSNR | 26.9770 | 26.9094 | **27.2508** | 30.9181 | 30.8306 | **30.9243** | 30.1568 | 30.0827 | **30.1977** |
| | SSIM | 0.8468 | 0.8460 | **0.8525** | 0.9377 | 0.9358 | **0.9377** | 0.9405 | 0.9397 | **0.9411** |
| RealBlur-J | PSNR | 26.1580 | 26.0855 | **26.3216** | 28.6370 | 28.5714 | **28.7067** | 28.9636 | 28.9273 | **28.9698** |
| | SSIM | 0.8094 | 0.8088 | **0.8127** | 0.8706 | 0.8649 | **0.8710** | 0.8792 | 0.8785 | **0.8796** |
| RealBlur-R | PSNR | 33.5516 | 33.4871 | **33.6879** | 35.9682 | 35.8307 | **36.0277** | 36.2017 | 36.1509 | **36.2136** |
| | SSIM | 0.9359 | 0.9349 | **0.9377** | 0.9472 | 0.9325 | **0.9476** | 0.9572 | 0.9563 | **0.9575** |
| | #Params | 11.72M | 11.46M | 10.51M | 15.74M | 13.89M | 13.33M | 25.31M | 25.06M | 24.92M |

and we split 485 for training and 15 for testing as the official selection. For the Huawei dataset, we randomly selected 2,200 images for training and kept the remaining 280 images for testing purposes. To evaluate the effectiveness of our proposed RubikConv, we compared it against two classical algorithms, namely SID [48] and DRBN [49], which served as baselines in our evaluation.

**Image De-blurring.** We evaluate RubikConv using the methodology of MPRNet [5] and Restormer [10] on the GoPro dataset [45], which includes 2,103 training pairs and 1,111 testing pairs. Furthermore, we evaluate the generalization capability of RubikConv on diverse datasets: HIDE [46] (synthetic), RealBlur-R [47] (real-world), and RealBlur-J [47] (real-world). The generalization test is performed using the GoPro-trained model. For comparison, we adopt three baselines: DeepDeblur [45] (classical), MPRNet [5], and Restormer [10] (latest image de-blurring techniques).

**Image Denoising.** Following [50, 10], we choose the popular SIDD dataset [51] as the training benchmark and evaluate the effectiveness of the proposed RubikConv on the test set of the SIDD dataset [51] and the DND dataset [52]. We employ three representative image denoising algorithms, DnCNN [33], MPRNet [5], and Restormer [10], as the baselines.

**Guided Image Super-resolution.** Following [53, 54], we evaluate the proposed RubikConv on the pan-sharpening task, a representative guided image super-resolution task. For comparison, we select a classical method (PanNet [55]) and two state-of-the-art algorithms (MutNet [56] and INNformer [57]) as baselines and evaluate on the WorldView II and GaoFen2 datasets [57, 58].

We employ widely-used image quality assessment metrics, including peak signal-to-noise ratio (PSNR), structural similarity index (SSIM), relative dimensionless global error in synthesis (ERGAS)[59], and spectral angle mapper (SAM)[60].

## 4.2 Implementation Details

Based on the above-mentioned competitive baselines, we create several variants of the baselines by replacing the standard convolution with the proposed Rubik's cube convolution:

1) **Original**: the baseline without any changes;

Table 3: Quantitative comparison of image denoising. The model is trained only on the SIDD [51] training set and directly tested on the SIDD [51] testing set and DND [52] dataset.

| Model | Metric | DnCNN | | | MPRNet | | | Restormer | | |
|---|---|---|---|---|---|---|---|---|---|---|
| | | Original | Conv1x1 | RubikConv | Original | Conv1x1 | RubikConv | Original | Conv1x1 | RubikConv |
| SIDD | PSNR | 37.1992 | 37.2247 | **37.4123** | 39.7129 | 39.6220 | **39.7741** | 40.0124 | 39.9429 | **40.0853** |
| | SSIM | 0.8954 | 0.8955 | **0.8986** | 0.9524 | 0.9502 | **0.9536** | 0.9587 | 0.9568 | **0.9593** |
| DND | PSNR | 38.3305 | 38.3946 | **38.5759** | 39.6276 | 39.5834 | 39.9043 | 40.0182 | 39.9885 | **40.0994** |
| | SSIM | 0.9307 | 0.9296 | **0.9328** | 0.9531 | 0.9524 | **0.9558** | 0.9560 | 0.9537 | **0.9564** |
| | #Params | 1.51M | 1.36M | 1.28M | 15.74M | 13.89M | 13.33M | 25.31M | 25.21M | 25.21M |

Table 4: Quantitative comparisons of guided image super-resolution.

| Model | Config | WorldView-II | | | | GaoFen2 | | | | #Params |
|---|---|---|---|---|---|---|---|---|---|---|
| | | PSNR↑ | SSIM↑ | SAM↓ | ERGAS↓ | PSNR↑ | SSIM↑ | SAM↓ | EGAS↓ | |
| PanNet | Original | 40.8212 | 0.9630 | 0.0257 | 1.0561 | 42.1699 | 0.9569 | 0.0192 | 0.9565 | 0.068M |
| | Conv1x1 | 40.8015 | 0.9624 | 0.0261 | 1.0576 | 42.1150 | 0.9527 | 0.0197 | 0.9576 | 0.061M |
| | RubikConv | **41.2692** | **0.9646** | **0.0248** | **1.0237** | **43.1764** | **0.9693** | **0.0176** | **0.8516** | 0.047M |
| MutNet | Original | 41.0692 | 0.9646 | 0.0248 | 1.0237 | 47.1699 | 0.9569 | 0.0192 | 0.9565 | 0.116M |
| | Conv1x1 | 40.9372 | 0.9624 | 0.02749 | 1.0072 | 47.1668 | 0.9563 | 0.0185 | 0.9576 | 0.115M |
| | RubikConv | **41.5258** | **0.9692** | **0.0230** | **0.9719** | **47.3274** | **0.9885** | **0.0103** | **0.5486** | 0.115M |
| INNFormer | Original | 41.5363 | 0.9696 | 0.0229 | 0.9649 | 47.1108 | 0.9882 | 0.0105 | 0.5610 | 0.061M |
| | Conv1x1 | 41.5276 | 0.9696 | **0.0226** | 0.9451 | 47.1295 | 0.9882 | 0.0104 | 0.5611 | 0.060M |
| | RubikConv | **41.5720** | **0.9699** | 0.0228 | **0.9649** | **47.2458** | **0.9886** | **0.0104** | **0.5609** | 0.060M |

2) **RubikConv**: replacing the standard convolution in the original model with our designed Rubik's cube convolution;

3) **Conv1x1**: a baseline that replaces the RubikConv in the setting of 2) with four convolution layers with $1 \times 1$ kernel for a fair comparison with approximately the same number of trainable parameters as 2).

For a fair comparison, we ensure each competitive baseline and its variants are subjected to the same training configuration and optimization strategy.

### 4.3 Comparison and Analysis

**Quantitative Comparison.** We conduct performance comparisons using the configurations defined above. The quantitative results for low-light image enhancement, image de-blurring, image denoising, and guided image super-resolution are presented in Tables 1 through 4, with the best results highlighted in bold. We observe a performance boost across all datasets and tasks when replacing the standard convolution layer with our proposed Rubik's cube convolution layer. In the case of the denoising and guided super-resolution tasks, our operator not only improves the performance of traditional methods like DnCNN and PanNet, but also helps push the performance of state-of-the-art methods like Restormer and INNFormer. Since Restormer has achieved the upper bound of the performance with a large model capacity, there are no significant performance gains for Restormer. Examining the results for SID/DRBN in Table 1, our Rubik's cube convolution achieves gains of 0.39dB/0.58dB over the original baseline on the LOL [43] dataset, while also improving generalizability on the Huawei [44] dataset with gains of 0.2dB/0.2dB. It also performs better than the "Conv1x1" variant, with gains of 0.52dB/0.51dB on the LOL dataset. These experiments demonstrate the potential of our Rubik's cube convolution in improving the performance of baseline models with minimal parameter overhead.

**Qualitative Comparison.** Due to the limited space, we only present the visual results of the low-light image enhancement task in Figure 5. It can clearly demonstrate that incorporating the RubikConv with the original baseline achieves more visually pleasing results. Specifically, in Figure 5 the model with the RubikConv achieves better lightness enhancement and color consistency with the corresponding normal-light images. More results can be found in the supplementary material.

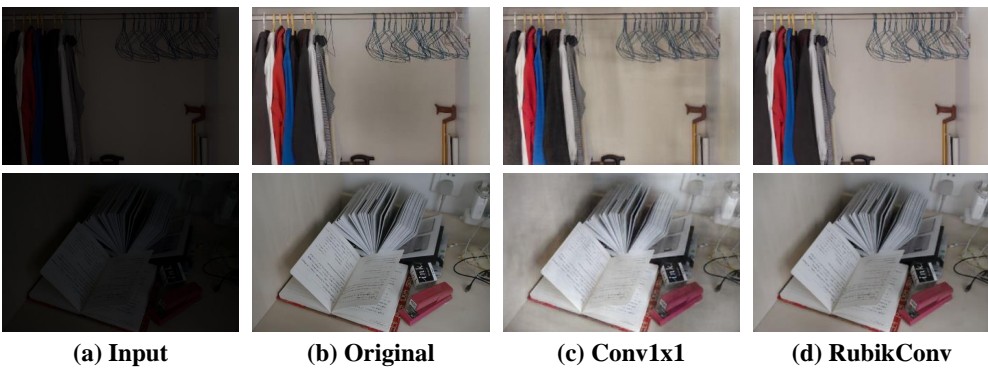

| (a) Input | (b) Original | (c) Conv1x1 | (d) RubikConv |

Figure 5: Visual comparison of DRBN [49] on the LOL [43] dataset.

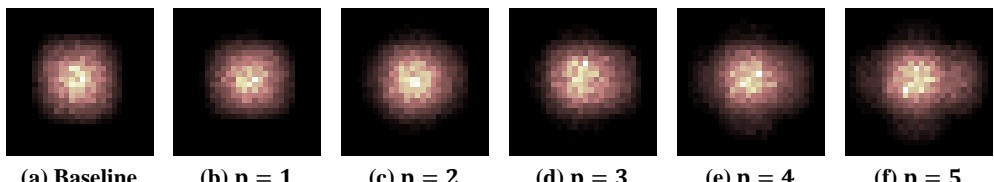

| (a) Baseline | (b) p = 1 | (c) p = 2 | (d) p = 3 | (e) p = 4 | (f) p = 5 |

Figure 6: The effective receptive field [42] of the DnCNN [33] with the proposed Rubik's cube convolution and p indicates the shifted pixels in the RubikConv. The results show that the effective receptive field of the network progressively expands as the number of shifted pixels increases.

**Expanding Receptive Field.** The vanilla convolution layer enables first-order channel interaction through linear channel-wise weighting, resulting in a consistent receptive field across the channel dimension. In contrast, our RubikConv generates a hierarchical receptive field through shifting and high-order channel-wise interactions. An interesting characteristic of RubikConv is that its receptive field expands as the number of shifted pixels increases. Figure 6 presents the effective receptive field [42] of the DnCNN [33] baseline, in comparison with our RubikConv across various pixel shift values. As can be observed, the baseline's effective receptive field approximates a square, while the RubikConv expands it in the left, right, up, and down directions. Moreover, as the number of shifted pixels increases, the overall size of the effective receptive field grows correspondingly.

## 4.4   Ablation Studies

To verify the robustness of shifting displacement and the ratio of identity, we perform the comparison over the following configurations:

1) **RubikConv-p**: replacing the standard convolution layer with the designed RubikConv where the number of the shifted pixel is "p";
2) **RubikConv-r**: substituting the standard convolution layer with our designed RubikConv, wherein the proportion of the unchanged identity component is denoted by "r".

We conduct ablation studies on the low-light image enhancement task with the SID [48] network and image denoising task with the DnCNN [33] network, and all the experiments with different configurations follow the same optimization strategy.

**Number of the Shifted Pixel.** To ascertain the influence of the number of shifted pixels in our proposed RubikConv, we conduct a series of experiments using the configuration denoted as 'RubikConv-p'. As observed from Table 5 and Figure 7, the performance remains consistent as the number of shifted pixels increases, up to a value of 3. Beyond this point, any further increase in p leads to a decline in performance, potentially due to the increased complexity in reconstructing the structure. Therefore, in all the experiments presented in this paper, we set p to 1 as the default value for the number of shifted pixels.

**Ratio of the Shifted Channel.** We investigate the impact of the ratio of the shifted channel following the definition in "RubikConv-r". Experimental results from both the SID dataset (as shown in Table 5) and the DnCNN network (as depicted in Figure 7) suggest that optimal performance is reached when

Table 5: Ablation studies of the low-light image enhancement network SID [48]. The model is trained on the LOL [43] training set and tested on the LOL [43] testing set and Huawei [44].

| Dataset | Metric | RubikConv-p | | | | | |
|---|---|---|---|---|---|---|---|
| | | Original | 1 (default) | 2 | 3 | 4 | 5 |
| LOL | PSNR | 20.1062 | 20.4972 | 20.5016 | 20.4858 | 20.4762 | 20.4506 |
| | SSIM | 0.7895 | 0.7979 | 0.7981 | 0.7976 | 0.7962 | 0.7954 |
| Huawei | PSNR | 16.5874 | 16.7867 | 16.7829 | 16.7913 | 16.7309 | 16.7288 |
| | SSIM | 0.5925 | 0.5941 | 0.5940 | 0.5940 | 0.5935 | 0.5934 |

| Dataset | Metric | RubikConv-r | | | | | |
|---|---|---|---|---|---|---|---|
| | | Original | 1/4 | 1/3 | 1/2 (default) | 2/3 | 3/4 |
| LOL | PSNR | 20.1062 | 20.4892 | 20.4860 | 20.4972 | 20.4992 | 20.4795 |
| | SSIM | 0.7895 | 0.7971 | 0.7969 | 0.7979 | 0.7961 | 0.0.7964 |
| Huawei | PSNR | 16.5874 | 16.7550 | 16.7548 | 16.7867 | 16.7782 | 16.7824 |
| | SSIM | 0.5925 | 0.5940 | 0.5937 | 0.5941 | 0.5937 | 0.5939 |

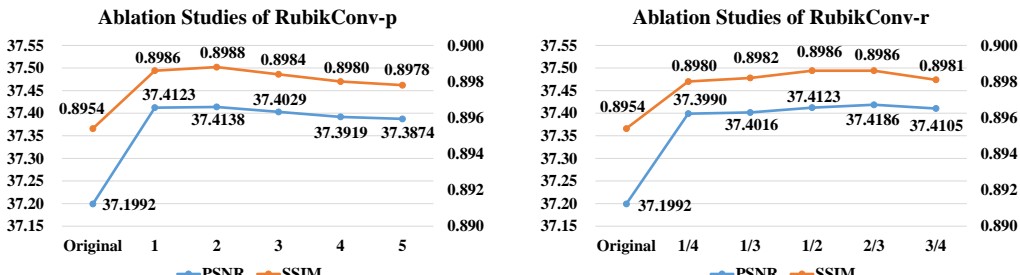

Figure 7: Ablation studies of the image denoising network DnCNN [33] on the SIDD [51] dataset.

the ratio is set to either 1/2 or 2/3. Further increases in the ratio r subsequently result in a decline in performance. We have set the default ratio r to 1/2 for all experiments conducted in this study.

## 5 Limitations

We will validate the effectiveness of the proposed Rubik's cube convolution on broader low-level tasks, such as image de-raining and de-hazing. As a generic operator, diverse comprehensive architectures integrated with our proposed operator should be explored. The primary aim of our research is beyond designing a universal operator for improving the performance of existing networks. We hope to offer an alternative tool for high-order channel-wise modeling. Consequently, our ongoing efforts will focus on showcasing its effectiveness across broader computer vision tasks.

## 6 Conclusion

In this paper, we study the modeling of relationships in the channel dimension of image restoration tasks, and propose a Rubik's cube convolution operator. It constructs the high-order channel-wise interactions and creates a Rubik's cube-like hierarchical receptive field in an efficient and effective manner. Note that it can be seamlessly incorporated into existing image restoration networks as a drop-in replacement for the standard convolution unit with fewer parameters. Extensive experiments across four distinct image restoration and enhancement tasks demonstrate the performance gains achieved by replacing our designed operator with the standard convolution uint.

## Broader Impact

Image restoration technology holds significant value across diverse fields, including remote sensing, medicine, astronomy, and consumer imaging equipment. Our proposed Rubik's cube convolution has the potential to enhance restoration algorithms by constructing high-order channel-wise relationships and producing a hierarchical receptive field along the channel dimension. This could lead to clearer, more detailed images that improve decision-making capabilities in these fields. While we anticipate no negative societal consequences from our work, future research and applications may reveal unforeseen impacts. Thus, the responsible use and ongoing evaluation of such technology is essential.

## Acknowledgements

This study is supported under the RIE2020 Industry Alignment Fund – Industry Collaboration Projects (IAF-ICP) Funding Initiative, as well as cash and in-kind contribution from the industry partner(s).

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
