# "Rubik's Cube: High-Order Channel Interactions with a Hierarchical Receptive Field" Supplementary Material

This supplementary document is organized as follows:

Section 1 illustrates the process of how to form a hierarchical receptive field within the combination of the shifting and interaction operations.

Section 2 provides the implementation details of Rubik's cube convolution within the image restoration baselines.

Section 3 provides the evaluation of our proposed Rubik's cube convolution on the classification task. We conduct the experiments on three widely-used classification benchmarks with three representative baselines.

Section 4 provides more quantitative and qualitative results.

## 1 The Hierarchical Receptive Field

As illustrated in Figure 1, given an input feature map $\mathbf{X} \in \mathbb{R}^{H \times W \times C}$, we evenly divide $\mathbf{X}$ into five parts by the channel dimension, where the first is kept unchanged and the remaining four ones are shifted in a distinct spatial direction: left, right, top, and down. Subsequent to the shifting operation, we discard out-of-focus pixels and any vacant pixels are filled with zeros. The shifted feature $\hat{\mathbf{X}}$ can be written as:

$$
\begin{aligned}
\hat{\mathbf{X}}[0:H, 0:W, 0:\mathbf{C}_{\mathrm{id}}] &\leftarrow \mathbf{X}[0:H, 0:W, 0:\mathbf{C}_{\mathrm{id}}], \\
\hat{\mathbf{X}}[0:H, 1:W, \mathbf{C}_{\mathrm{id}}:\mathbf{C}_{\mathrm{id}}+\mathbf{C}_{\mathrm{g}}] &\leftarrow \mathbf{X}[0:H, 0:W-1, \mathbf{C}_{\mathrm{id}}:\mathbf{C}_{\mathrm{id}}+\mathbf{C}_{\mathrm{g}}], \\
\hat{\mathbf{X}}[0:H, 0:W-1, \mathbf{C}_{\mathrm{id}}+\mathbf{C}_{\mathrm{g}}:\mathbf{C}_{\mathrm{id}}+2\mathbf{C}_{\mathrm{g}}] &\leftarrow \mathbf{X}[0:H, 1:W, \mathbf{C}_{\mathrm{id}}+\mathbf{C}_{\mathrm{g}}:\mathbf{C}_{\mathrm{id}}+2\mathbf{C}_{\mathrm{g}}], \\
\hat{\mathbf{X}}[0:H-1, 0:W, \mathbf{C}_{\mathrm{id}}+2\mathbf{C}_{\mathrm{g}}:\mathbf{C}_{\mathrm{id}}+3\mathbf{C}_{\mathrm{g}}] &\leftarrow \mathbf{X}[1:H, 0:W, \mathbf{C}_{\mathrm{id}}+2\mathbf{C}_{\mathrm{g}}:\mathbf{C}_{\mathrm{id}}+3\mathbf{C}_{\mathrm{g}}], \\
\hat{\mathbf{X}}[1:H, 0:W, \mathbf{C}_{\mathrm{id}}+3\mathbf{C}_{\mathrm{g}}:\mathbf{C}_{\mathrm{id}}+4\mathbf{C}_{\mathrm{g}}] &\leftarrow \mathbf{X}[0:H-1, 0:W, \mathbf{C}_{\mathrm{id}}+3\mathbf{C}_{\mathrm{g}}:\mathbf{C}_{\mathrm{id}}+4\mathbf{C}_{\mathrm{g}}],
\end{aligned}
\tag{1}
$$

where $\mathbf{C}_{\mathrm{id}}$ is the number of channels of the unchanged identity part, $\mathbf{C}_{\mathrm{g}}$ is the number of channels of a shifted group, and $\mathbf{C}_{\mathrm{id}} + 4 * \mathbf{C}_{\mathrm{g}} = \mathbf{C}$. Next, the shifted feature $\hat{\mathbf{X}}$ is split into $\hat{\mathbf{X}}_{\mathrm{ori}} \in \mathbb{R}^{H \times W \times C_{\mathrm{id}}}$ and $\{\hat{\mathbf{X}}_{c1}, \hat{\mathbf{X}}_{c2}, \hat{\mathbf{X}}_{c3}, \hat{\mathbf{X}}_{c4}\} \in \mathbb{R}^{H \times W \times C_{\mathrm{g}}}$ along the channel dimension.

After the shifting operation, we construct the high-order interaction in the channel dimension as described in the manuscript. As shown in Figure 1, for the $(i, j)$ pixel in the up-shifting group, $\hat{\mathbf{X}}_{c1}$, it will interacts with the $(i + p, j)$ pixel in the down-shifting group, $\hat{\mathbf{X}}_{c12}$, and $p$ indicates the number of the shifted pixels. Therefore, the combination of the shifting and interaction leads to a stratified receptive field along the channel dimension, reminiscent of a Rubik's cube.

Submitted to 37th Conference on Neural Information Processing Systems (NeurIPS 2023). Do not distribute.

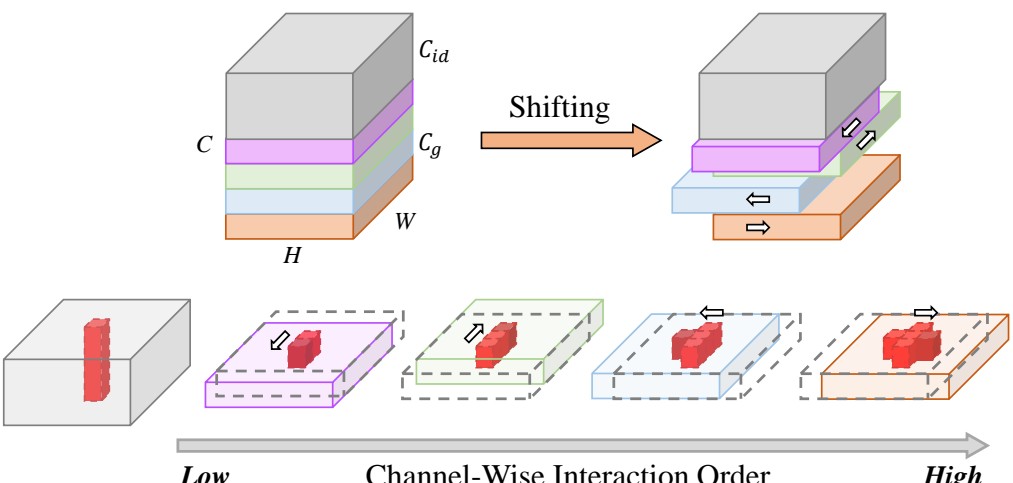

Figure 1: An illustration of the shifting operation and the formulation of the hierarchical receptive field. Specifically, the input feature is separated into five groups, where the last four are shifted into four direction and the first is unchanged. When a up-shifting group interacts the next down-shifting group, the $(i, j)$ pixel will interweave with its neighboring pixel, $(i + p, j)$, where $p$ denotes the number of shifted pixels. Therefore, with the combined action of the shifting and interaction operation, the receptive field will be expanded along the downward direction. The red-shaded region indicates the receptive field and the scarlet-shaded region presents the newly expanded receptive field after the corresponding interaction.

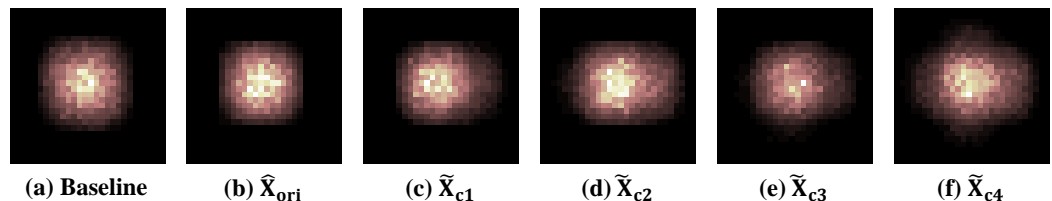

| (a) Baseline | (b) $\widehat{\mathbf{X}}_{\mathrm{ori}}$ | (c) $\widetilde{\mathbf{X}}_{\mathrm{c1}}$ | (d) $\widetilde{\mathbf{X}}_{\mathrm{c2}}$ | (e) $\widetilde{\mathbf{X}}_{\mathrm{c3}}$ | (f) $\widetilde{\mathbf{X}}_{\mathrm{c4}}$ |

Figure 2: **Visualization of the effective receptive field [1].** (a) The baseline indicates the effective receptive field of the default architecture, and the last five describe the effective receptive field of the (b) identity group, $\widehat{\mathbf{X}}_{\mathrm{ori}}$, (c) left-shifting group, $\widetilde{\mathbf{X}}_{\mathrm{c1}}$, (d) right-shifting group, $\widetilde{\mathbf{X}}_{\mathrm{c2}}$, (e) up-shifting group, $\widetilde{\mathbf{X}}_{\mathrm{c3}}$, (f) and down-shifting group, $\widetilde{\mathbf{X}}_{\mathrm{c4}}$ after interactions in the Rubik's cube convolution.

We visualize the effective receptive field [1] of the distinct groups in the Rubik's cube convolution. Figure 2 demonstrates the effective receptive field expands along the up, down, left, and right directions after the corresponding shifting and interaction.

## 2 Implementation Details

Based on the competitive baselines, we create several variants of the baselines by replacing the standard convolution with the proposed Rubik's cube convolution:

1) **Original**: the baseline without any changes;
2) **RubikConv**: replacing the standard convolution in the original model with our designed Rubik's cube convolution;
3) **Conv1x1**: a baseline that replaces the RubikConv in the setting of 2) with four convolution layers with $1 \times 1$ kernel for a fair comparison with approximately the same number of trainable parameters as 2).

Taking a network consisted of a stack of convolution layers with a $3 \times 3$ kernel for example, we replace the standard convolution layer with our proposed Rubik's cube convolution layer in 2) or four convolution with $1 \times 1$ kernel in 3) from top to bottom.

# 3 Evaluation on Image Classification

Due to the limited space in the main manuscript, we provide the evaluation on the image classification task in the supplementary material. We choose three widely-used image classification benchmarks: CIFAR-10 [2], CIFAR-100 [2], and CUB [3]. For comparison, we employ three algorithms: AlexNet [4], VGG [5], and ResNet [6].

To validate the effectiveness of the proposed approach, we conduct extensive experiments as described in the implementation details. The quantitative results are presented in Table 1, where the best results are highlighted in bold. For the three recognition models across three benchmarks, integrating our Rubik's cube convolution operation into the baseline will achieve the performance improvement, demonstrating the effectiveness of our operation in the recognition task.

Table 1: Quantitative comparison of image classification. "Acc-1" and "Acc-5" indicate the top-1 and top-5 classification accuarcy.

| Model | Metric | CIFAR-10 | | | CIFAR-100 | | | CUB | | |
|---|---|---|---|---|---|---|---|---|---|---|
| | | Original | Conv1x1 | RubikConv | Original | Conv1x1 | RubikConv | Original | Conv1x1 | RubikConv |
| AlexNet | Acc-1 | 77.98 | 75.48 | **82.63** | 49.98 | 48.27 | **52.77** | 61.99 | 60.05 | **64.31** |
| | Acc-5 | 98.69 | 95.71 | **98.94** | 70.43 | 66.41 | **76.12** | 83.48 | 83.09 | **86.73** |
| VGG-16 | Acc-1 | 84.62 | 83.54 | **87.40** | 55.76 | 55.06 | **57.39** | 79.70 | 78.52 | **80.79** |
| | Acc-5 | 99.22 | 96.09 | **99.26** | 76.36 | 76.43 | **79.20** | 88.34 | 86.85 | **89.06** |
| ResNet-18 | Acc-1 | 89.45 | 87.85 | **91.90** | 59.63 | 58.74 | **62.06** | 85.72 | 84.16 | **87.05** |
| | Acc-5 | 99.65 | 96.83 | **99.70** | 82.08 | 80.47 | **83.90** | 93.60 | 92.87 | **94.65** |

# 4 Experiments

**Quantitative Comparison.** Due to the limited space, we present the comparison on the World-III dataset in the supplementary material. As the experiments on the manuscript, we adopt three baselines (PanNet [7], MulNet [8], and INNFormer [9]) for evaluation. We conduct experiments as described in the implementation details. As described in Table 2 , we observe a performance gain by integrating our proposed Rubik's cube convolution across all competitive baselines.

Table 2: **Quantitative comparisons of pan-sharpening.**

| Model | Configurations | WorldView-III | | | |
|---|---|---|---|---|---|
| | | PSNR↑ | SSIM↑ | SAM↓ | ERGAS↓ |
| PanNet | Original | 29.6863 | 0.9072 | 0.0853 | 3.4260 |
| | Conv1x1 | 29.4305 | 0.8973 | 0.1008 | 3.6954 |
| | RubikConv | **39.9831** | **0.9139** | **0.0812** | **3.2453** |
| MulNet | Original | 30.4807 | 0.9211 | 0.0769 | 3.1196 |
| | Conv1x1 | 30.4186 | 0.9131 | 0.0837 | 3.3364 |
| | RubikConv | **30.6426** | **0.9231** | **0.0.0754** | **3.0835** |
| INNFormer | Original | 30.4349 | 0.9204 | 0.0756 | 3.1439 |
| | Conv1x1 | 30.3850 | 0.9175 | 0.0827 | 3.3061 |
| | RubikConv | **30.5052** | **0.9211** | **0.0742** | **3.1118** |

**Qualitative Comparison.** Due to the limited space in the manuscript, we present more visualizations in the supplementary materials. As illustrated in Figure 3 and 4, integrating our Rubik's cube convolution into baselines generate the visually pleasant enhanced results. Specifically, the baseline and the baseline with Conv1x1 fails to recover texture and suffers from artifacts and color distortion. In contrast, the baseline combined with our Rubik's cube convolution operator achieves details reconstruction, artifact reduction, and color consistency.

We also provide the visual comparison on the image de-noising task in Figure 5 and 6. The qualitative results consistently demonstrate that the improvement on the visual quality by integrating our Rubik's cube convolution. The comprehensive visual results in the supplementary material demonstrate the effectiveness of our proposed operator.

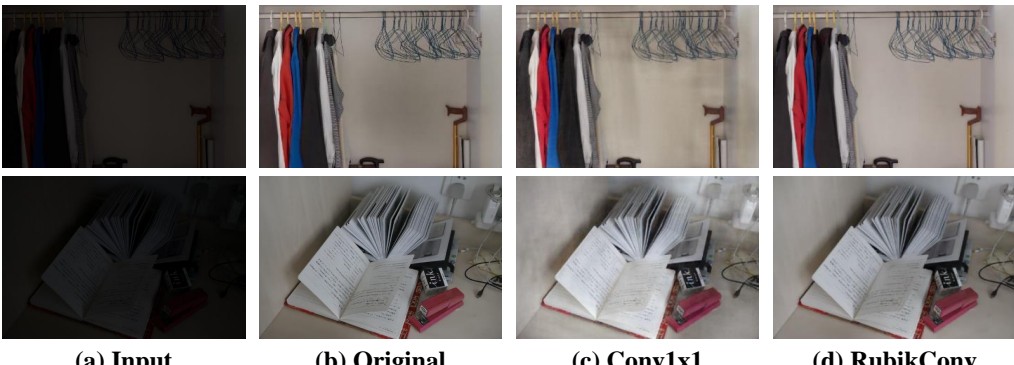

| (a) Input | (b) Original | (c) Conv1x1 | (d) RubikConv |

Figure 3: Visual comparison of DRBN [10] on the LOL [11] dataset.

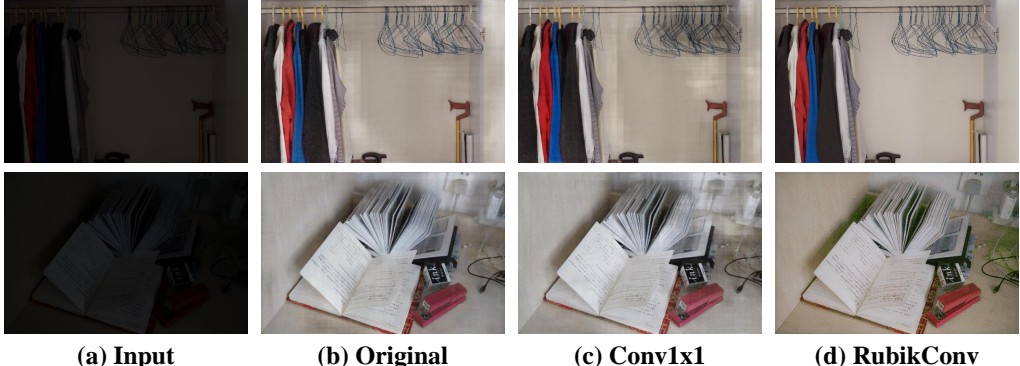

| (a) Input | (b) Original | (c) Conv1x1 | (d) RubikConv |

Figure 4: Visual comparison of SID [12] on the LOL [11] dataset.

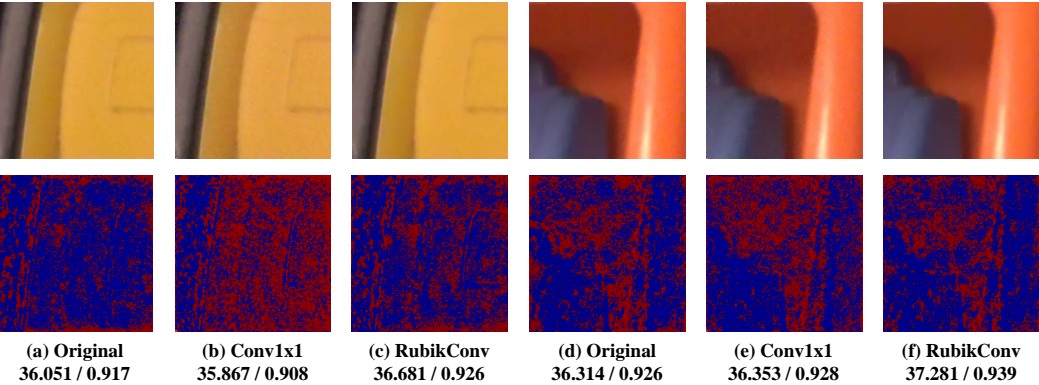

| (a) Original | (b) Conv1x1 | (c) RubikConv | (d) Original | (e) Conv1x1 | (f) RubikConv |
| 36.051 / 0.917 | 35.867 / 0.908 | 36.681 / 0.926 | 36.314 / 0.926 | 36.353 / 0.928 | 37.281 / 0.939 |

Figure 5: Visual comparison of DnCNN [13] on the SIDD [14] dataset. The second row presents the error map between the corresponding denoised results and the clean images. The numbers indicate the corresponding PSNR/SSIM metrics.

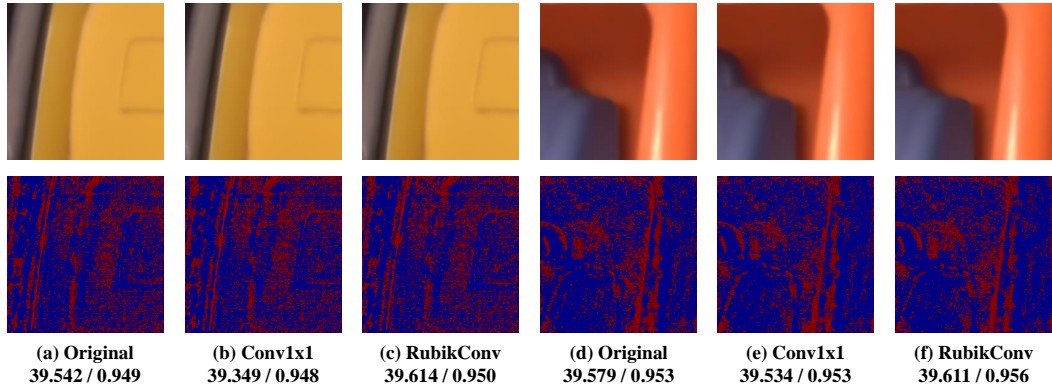

| (a) Original
39.542 / 0.949 | (b) Conv1x1
39.349 / 0.948 | (c) RubikConv
39.614 / 0.950 | (d) Original
39.579 / 0.953 | (e) Conv1x1
39.534 / 0.953 | (f) RubikConv
39.611 / 0.956 |

Figure 6: Visual comparison of MPRNet [15] on the SIDD [14] dataset. The second row presents the error map between the corresponding denoised results and the clean images. The numbers indicate the corresponding PSNR/SSIM metrics.