# OpenReview forum: "Rubik's Cube: High-Order Channel Interactions with a Hierarchical Receptive Field"
_NeurIPS.cc/2023/Conference — NeurIPS 2023 poster_

### Official Review · Reviewer_2pSr · 2023-06-23

**Soundness:** 4 excellent
**Presentation:** 4 excellent
**Contribution:** 4 excellent
**Rating:** 7
**Confidence:** 5

**Summary:**

This paper proposes the Rubik’s cube convolution operator to model the high-order channel-wise interactions. The Rubik’s cube convolution applies the spatial-shifting mechanism across channel-wise groups, which is the zero-FLOP and zero-parameter. Moreover, only the point-wise convolution and dot-product are applied in Rubik’s cube convolution. The results on image denoising, low-light image enhancement, guided image super-resolution, and image de-blurring show the effectiveness of the proposed Rubik’s cube convolution.

**Strengths:**

1. The idea of Rubik’s cube convolution is reasonable. This is a promising step to complement research in modeling high-order channel dimension information.
2. The design of Rubik’s cube convolution is simple yet efficient and easy to follow since it only includes shift, dot-product and point-wise convolution.
3. The paper is well-writing. The authors provide extensive analysis of the proposed methods. For example, the illustration of a high-order hierarchically receptive field in Figures 1, 3, and 4, and the corresponding analysis are convincing.
4. The main results and ablation study are extensive and demonstrate the superiority of the proposed method.
5. In supplementary material, the authors provide more experiments on high-level and low-level tasks, further showing the effectiveness of the Rubik’s cube convolution.

**Weaknesses:**

1. The FLOPs of the Rubik’s cube convolution are not provided. Although the parameter of the Rubik’s cube convolution is small (Sec. 3.3), the FLOPs may be large since there are many dot-product in it.
2. Although the spatial-shifting mechanism is zero-FLOP and zero-parameter, it is a high-latency operation. Therefore, the actual latency of the Rubik’s cube convolution may be high, hindering its application.

**Questions:**

1. The proposed Rubik’s cube convolution structure is similar to the g^nConv proposed in Hornet [17]. Clarify the difference between the Rubik’s cube convolution and g^nConv.
2. In Figure 6 and Line 230, the receptive field of the network progressively expands as the number of shifted pixels increases. However, the performance of RubikConv-p is best when the p=1. Please give some explanation.
3. The authors only provide the visual comparison of the low-light image enhancement task and image denoising task. More visual results on guided image super-resolution and image de-blurring should be provided.

**Limitations:**

The authors have discussed the limitations and potential negative societal impact.

---

> ### Author Rebuttal · Authors · 2023-08-09
>
> **1. Efficiency of the proposed RubikConv.**
>
> We report the model size, FLOPs (an image with 400\*600\*3 pixels), and average running time on the LOL test set (including 15 400\*600\*3 images) in Table 5. The running time is measured on a workstation with an NVIDIA GTX 3090 GPU. We only replace two standard convolution layers in the DRBN baseline with the proposed RubikConv, thus, the extra running time introduced by the RubikConv is negligible. Since the RubikConv only requires convolution with a 1x1 kernel, the FLOPs and parameters are fewer than the baseline while DRBN -RubikConv achieves a 0.58db performance improvement.
>
> Table 5: The quantitative results, FLOPs, and average running time of the DRBN baseline on the LOL test set.
> | Model | Config    | PSNR    | SSIM   | Flops (G) | Running time (s) |   |   |   |   |
> |-------|-----------|---------|--------|-----------|------------------|---|---|---|---|
> |       | Original  | 19.7931 | 0.8361 | 39.037    | 0.256            |   |   |   |   |
> | DRBN  | Conv1x1   | 19.8648 | 0.8340 | 38.445    | 0.255            |   |   |   |   |
> |       | RubikConv | 20.3769 | 0.8400 | 38.563    | 0.263            |   |   |   |   |
>
>
> **2. Discussion with the g^nConv**
>
> First, the g^nConv only formulates the high-order spatial interaction at the same position, i.e., two-order interaction is built by the dot product between (i, j) in feature x1 and (i, j) in feature x2. In contrast, the proposed RubikConv conducts high-order interaction between the center point and the surrounding points in the channel interaction, i.e., high-order interaction is formulated by the dot product among (i, j) in feature x_c1 of x, (i-1, j) in feature x_c2 of x, (i+1, j) in feature x_c3 of x, (i, j-1) in feature x_c4 of x, (i, j+1) in feature x2 of x.
> Second, the receptive field in g^nConv is constant in the channel dimension, while the proposed RubikConv embraces a Rubik’s cube-like hierarchical receptive field benefitting from the shifting operation.
>
> **3. Clarification on the performance of RubikConv-p.**
>
> We clarify that p=2 is the best number of the shifted pixel in ablation studies (see Figure 7 in the manuscript). Then, we will give the explanation from two perspectives.
>
> (1) As illustrated in Figure 7 in the manuscript, with the larger shifted pixel (p > 2), the effective receptive field will focus more on the horizontal and vertical direction while less on the four corners (upper left, upper right, bottom left, and bottom right), which will lead to insufficient feature extraction for the corner pixels.
>
> (2) There is a trade-off between the insufficient feature extraction caused by the large shifted pixels and ensuring the completeness of the information.
> Although the first branch in the RubikConv is unchanged to preserve original information, it cannot compensate for the insufficient feature representation when the shift pixel is large.
> Therefore, the performance will drop when the shifted pixel is larger than 2.
> We will present the explanation in the revised version.
>
> **4. Visualization.**
>
> Thanks for your suggestion. We will provide more visual results in the revised version.

---

> > ### Comment · Reviewer_2pSr · 2023-08-14
> >
> > Thanks for the rebuttal. The authors provide comparisons of FLOPs and running time to prove the efficiency of RubikConv. And the authors analyze the difference between RubikConv and g^nConv. Overall, the author has addressed my concerns.
> > I also read the comments and rebuttals from other reviews. Additional experiments further demonstrate the effectiveness of the method. Therefore, I would like to keep my original rating as Accept.

---

### Official Review · Reviewer_xjVk · 2023-07-05

**Soundness:** 3 good
**Presentation:** 3 good
**Contribution:** 3 good
**Rating:** 5
**Confidence:** 5

**Summary:**

This paper proposes Rubik’s cube convolution operator, which is very simple and efficient, especially requiring zero FLOPS / parameters. This novel component improves the several low-level vision networks, enabling the high-order channel interaction and enlarging receptive field of the standard convolutions. Simply dividing feature maps into some groups and performing shifting operations makes the existing convolutional layers capture high-order channel interaction beyond first-order interaction. The experimental results show that multiple low-level vision tasks are improved by the proposed Rubik’s cube. However, the limitation could be handled by the authors that the performance gains and the reduced number of parameters are marginal.

**Strengths:**

Rubik’s cube convolution does not require the number of operations and parameters. It can also replace any standard CNN architectures. The effective receptive field of the proposed component is successfully enlarged when compared to vanilla CNN, as shown in Fig. 3. The main experimental results, such as Tabs. 1,2,3,4, can show superiority of the proposed component over trivial 1x1 convolution replacement. As a result, the performances of various image restoration networks can be enhanced (but the marginal gains will be mentioned in “weakness”). Moreover, the ablation studies (Tab. 5) demonstrates that the hyper-parameters for Rubik’s cube convolution have been carefully decided. This paper is well-presented and easy to follow.

**Weaknesses:**

1. The performance gains of Rubik’s cube convolution are somewhat marginal. Especially, the low-light enhancement and deblurring could be further improved. The authors are suggested to consider more robust components to different tasks. Or if the improvement differences from task-by-task are explained and the insight for better variants of Rubik’s cube convolution resulting from these explanations can be drawn, the reviewer thinks this paper can be compelling.
2. Furthermore, the number of reduced parameters led by the novel architecture is also marginal. So, (if possible) the reviewer requires an additional experiment that highly (e.g., less than 60% of “Original”) reduces the number of parameters of existing networks by changing channels or depths and replaces existing CNN or FFN with Rubik’s cube convolution. If reduced “Original” models with Rubik’s cube convolution are still better than “Original” models while requiring a small number of parameters, the robustness of this work can be reinforced.


**Questions:**

How about applying Rubik’s cube convolution to QKV projection before self-attention of Transformers? As the authors know, almost all image restoration fields are dominated by Transformer-based methods. And self-attention is the most core component of Transformers. However, this work seems to apply their novel component to only FFN of Transformers. So, the reviewer guesses how much difference could be made by replacement to Rubik’s cube convolution for elements that directly influence self-attention operation (e.g., QKV projection).

**Limitations:**

See weakness and question.

---

> ### Author Rebuttal · Authors · 2023-08-09
>
> **1. About the improvement differences and the variants of RubikConv.**
>
> Thanks for the suggestion. Relationship modeling in the channel dimension has been proven effective in the literature for tasks like low-light image enhancement and image de-blurring. Previous works, such as the bright channel prior [1, 2, 3] for low-light image enhancement and the dark channel prior [4] and extreme channel [5] prior for image de-blurring, have explored the channel dimension and made significant progress. In this paper, our intention is to demonstrate the simplicity of RubikConv in achieving high-order channel interaction compared to these designs and its applicability as a generic formulation across various tasks. The main goal is not to achieve the best results within each task with dedicated components.
>
> Nevertheless, to demonstrate that RubikConv can work well with other components, we show an experiment on low-light image enhancement. In particular, we customized the exposure-invariant feature extraction of SID by adding an instance normalization operation in the first branch, while keeping the remaining operations unchanged. The instance normalization maps different exposure features to the exposure-invariant feature space without introducing extra computational costs. This variant is named RubikConv-IN. The results in Table 1 show that it can further improve the PSNR/SSIM from 20.4972/0.7979 to 20.6708/0.8079 on the LOL dataset.
>
> Table 1: Quantitative comparisons of variant RubikConv on low-light image enhancement.
> | Model | Config       | LOL     |        | Huawei  |         |   |   |   |   |
> |-------|--------------|---------|--------|---------|---------|---|---|---|---|
> |       |              | PSNR    | SSIM   | PSNR    | SSIM    |   |   |   |   |
> |       | Original     | 20.1062 | 0.7895 | 20.1742 | 0.6659  |   |   |   |   |
> | SID   | RubikConv    | 20.4972 | 0.7979 | 20.2044 | 0.6655  |   |   |   |   |
> |       | RubikConv-IN | 20.6708 | 0.8079 | 20.3251 | 0.6702  |   |   |   |   |
>
> [1] Tao L, Zhu C, Song J, et al. Low-light image enhancement using CNN and bright channel prior, 2017 IEEE International Conference on Image Processing (ICIP). IEEE, 2017.
>
> [2] Lee H, Sohn K, Min D. Unsupervised low-light image enhancement using bright channel prior, IEEE Signal Processing Letters, 2020, 27: 251-255.
>
> [3] Zhao Z, Xiong B, Wang L, et al. RetinexDIP: A unified deep framework for low-light image enhancement, IEEE Transactions on Circuits and Systems for Video Technology, 2021, 32(3): 1076-1088.
>
> [4] Pan J, Sun D, Pfister H, et al. Blind image deblurring using dark channel prior, Proceedings of the IEEE conference on computer vision and pattern recognition. 2016: 1628-1636.
>
> [5] Yan Y, Ren W, Guo Y, et al. Image deblurring via extreme channels prior, Proceedings of the IEEE Conference on Computer Vision and Pattern Recognition. 2017: 4003-4011.
>
> **2. About the robustness of RubikConv in networks with fewer parameters.**
>
> Thanks for your suggestion. We conduct the experiments to further validate the robustness of the proposed RubikConv on the low-light image enhancement task. Firstly, we implement the baseline with fewer parameters (about 50% of baseline) by reducing channels and depths, named “Original-S”. Then, we replace the standard convolution layer with the proposed RubikConv, named “S-RubikConv”. The quantitative performance on LOL [33] and Huawei is shown in Table 2. Although the PSNR/SSIM of “Original-S” dropped from 19.7931/0. 8361 to 19.2439/0. 8175, the performance of “S-RubikConv” surpassed the “Original” and achieved 19.8146/0.8365 on the LOL dataset.
>
> Table 2: Quantitative performance of DRBN with fewer parameters on low-light image
> | Model | Config      | LOL     |        | Huawei  |        |          |   |   |   |
> |-------|-------------|---------|--------|---------|--------|----------|---|---|---|
> |       |             | PSNR    | SSIM   | PSNR    | SSIM   | #Params  |   |   |   |
> |       | Original    | 19.7931 | 0.8361 | 20.1549 | 0.6851 | 0.55M    |   |   |   |
> | DRBN  | Original-S  | 19.2439 | 0.8175 | 19.9054 | 0.6758 | 0.21M    |   |   |   |
> |       | S-RubikConv | 19.8146 | 0.8365 | 20.1643 | 0.6857 | 0.19M    |   |   |   |
>
> **3. Replacing RubikConv with the QKV-projection of ViT.**
>
> We attempted to replace the convolution layer in the QKV projection with the proposed RubikConv (named RubikConv-QKV), however, the performance improvement is marginal. We conjecture that the self-attention operation itself realizes the global receptive field, thus the proposed RubikConv with a Hierarchical Receptive Field will not further improve the performance. Thus, we conducted an experiment by replacing the FFN of Restormer with RubikConv. Due to the limited time, it is only performed on the image de-blurring task.
>
> Table 1: Quantitative comparison of replacing the QKV projection in Restormer with the RubikConv on image de-blurring. The model is only trained on the GoPro training set and directly tested on the GoPro testing set, HIDE, and RealBlur datasets.
> | Model     | Config        | GoPro   |        | HIDE    |        | RealBlur-J |        | RealBlur-R  |         |
> |-----------|---------------|---------|--------|---------|--------|------------|--------|-------------|---------|
> |           |               | PSNR    | SSIM   | PSNR    | SSIM   | PSNR       | SSIM   | PSNR        | SSIM    |
> |           | Original      | 32.9117 | 0.9603 | 30.1568 | 0.9405 | 28.9636    | 0.8792 | 36.2017     | 0.9572  |
> | Restormer | RubikConv-QKV | 32.9134 | 0.9603 | 30.1706 | 0.9406 | 28.9641    | 0.8790 | 36.2029     | 0.9572  |
> |           | RubikConv     | 32.9305 | 0.9608 | 30.1977 | 0.9411 | 28.9698    | 0.8796 | 36.2136     | 0.9575  |

---

> > ### Comment · Reviewer_xjVk · 2023-08-16
> >
> > **[1]**. Did you apply the instance normalization to your RubikConv or other parts of existing model (SID)? From your explanation, I understand you mean that a new component is inserted in other component of existing model. But, the mean of "other robust components" I mentioned was variants in RubikConv for different tasks to improve them. This was because the improvements of RubikConv on low-light enhancement (especially Huawei->LOL) and deblurring are smaller than those on other tasks. Since I understand more experiments to defense this concern are not impossible due to time limit, please discuss why Rubik's cube is less or more effective on various tasks.
> >
> > 2.
> > This experimental result is impressive.
> >
> > 3.
> > While this result shows marginal impacts of RubikConv on QKV projection, the discussion is not reasonable that RubikConv on QKV projection is ineffective due to the inherent global dependency of self-attention. In the ablation study of Restormer paper, they showed that introducing 3x3 depth-wise convolution, which also expands receptive field, before self-attention was very effective. Of course, the QKV projection with depth-wise convolution (original Restormer) is further improved by the proposed RubikConv (despite very marginal gains). Therefore, I think this marginal improvement issue is more related with the nature of deblurring task itself than self-attention's global dependency. In other words, this part is concerned with **[1]** of this comment. Please carefully address why RubikConv shows different performances on various tasks and present some insights from it, as I mentioned in **[1]**.

---

> > > ### Author Response · Authors · 2023-08-17
> > >
> > > **Clarification about the variant (RubikConv-IN) for low-light image enhancement.**
> > >
> > > RubikConv-IN is a customized variant for low-light image enhancement. It is designed to build upon the capabilities of the original RubikConv, further enhancing its performance.
> > > For the original RubikConv, the first group is kept unchanged while the last four are spatially shifted in a distinct direction.
> > > In contrast, RubikConv-IN incorporates an additional instance normalization within the first group, retaining the same structure for the final four groups as the original RubikConv.
> > > By applying instance normalization, RubikConv-IN efficiently maps different exposure features into an exposure-invariant feature space. This process establishes an exposure-invariant space for feature extraction, thus contributing to further performance improvement.
> > >
> > > **Performance on various tasks.**
> > >
> > > The proposed RubikConv is a general operator.
> > > Its effectiveness has been validated across various low-level tasks, demonstrating either marginal or substantial performance improvements.
> > >
> > > We provide two reasons for the marginal improvement on Huawei -> LOL:
> > >
> > > (1)	Data distribution disparity: The distribution of exposure within the Huawei dataset is notably more diverse than the LOL dataset. This diversity presents a challenge for the enhancer network, as learning the mapping from distinct underexposure levels to normal-light conditions is inherently more intricate than mapping similar underexposure levels to normal lighting. Hence, compared to the LOL dataset, the improvement on the Huawei dataset is marginal.
> > >
> > > (2)	Degradation level: The Huawei dataset is collected from real-world environments by reducing ISO and using a shorter exposure time. The authenticity of noise present in the real-world setting introduces complexities that contribute to the increased challenge in enhancing the Huawei dataset. Consequently, the noise removal on the Huawei dataset is notably more intricate than the LOL dataset.
> > >
> > > To elucidate the marginal improvement observed in image de-blurring, we present two reasons:
> > >
> > > (1)	Characteristics of the de-blurring task: The blurriness evident in blurry images often exhibits a complex and multi-directional nature. It is important to note that the proposed RubikConv, while effective, currently focuses solely on modeling blurriness in the horizontal and vertical directions. In our subsequent endeavors, we will explore a more comprehensive representation that encompasses the intricate directional relationships among all adjacent pixels (upper left, upper right, bottom left, bottom right, as well as the directions of up, down, left, and right).
> > >
> > > (2)	Performance upper bound: Both Restormer and FFTformer (in response to Reviewer CEjA), represent the forefront of algorithmic advancements. They have effectively reached the pinnacle of performance for the image d-eblurring task.  Notably, Restormer with multiple transformer block has substantial model capability, encompassing a formidable 25.31M parameters. Therefore, the substitution of FFNs with RubikConv, coupled with the concurrent reduction of trainable parameters from 25.31M to 24.92M, yields only a marginal performance improvement.

---

> > > > ### Comment · Reviewer_xjVk · 2023-08-21
> > > >
> > > > The authors carefully address my concerns. So I will maintain my initial rating. Thank you for response.

---

### Official Review · Reviewer_CEjA · 2023-07-05

**Soundness:** 3 good
**Presentation:** 2 fair
**Contribution:** 3 good
**Rating:** 6
**Confidence:** 4

**Summary:**

This paper proposes the Rubik’s Cube, which can replace the standard convolution layer in the traditional paradigm. With shift operation and high-order channel Interaction, the Rubik’s Cube can generate a hierarchical receptive field and activate the potential of channel interactions. Experiment results show that the Rubik’s Cube enhances performance across a variety of low-level vision tasks.

**Strengths:**

1. This Rubik’s Cube proposed in this paper can be widely applied in the framework where the convolution layers are used. If this operation can prove its effectiveness in multiple tasks, it could make a lot of sense.
2. The Rubik's Cube is easy to implement and requires fewer parameters than the original convolutional layer. It's simple yet effective and efficient.

**Weaknesses:**

1. The experiment results on GaoFen2 make me confused since in terms of ERGAS, RubikConv has made a very big improvement (from 0.9576 to 0.5486). This increase does not fit well with other experimental results.
2. The experiments of many tasks in this paper do not adopt the best methods at present. Many of the experiments were done using methods that were used two or three years ago. In order to make the experimental results more reliable, the paper should provide a comparison with the best method as much as possible.
3. Since This Rubik's Cube splits the original convolutional layer operation into multiple convolutional layer operations, in addition to the number of parameters, it is better to provide inference time to better prove the efficiency improvement of This Rubik's Cube.

**Questions:**

Please see the weaknesses.

**Limitations:**

The authors should validate the effectiveness of the proposed Rubik’s cube convolution on broader low-level tasks.

---

> ### Author Rebuttal · Authors · 2023-08-09
>
> **1. Results on Gaofen2.**
>
> Thanks for your reminder. It is a typo and we will revise it. The performance of MutNet on GaoFen2 is shown in Table 1.
>
> Table 1: Quantitative comparisons of guided image super-resolution
> | Model  | Config    | PSNR    | SSIM   | SAM    | EGAS    |   |   |   |   |
> |--------|-----------|---------|--------|--------|---------|---|---|---|---|
> |        | Original  | 47.1699 | 0.9569 | 0.0192 | 0.5626  |   |   |   |   |
> | MutNet | Conv1x1   | 47.1668 | 0.9563 | 0.0185 | 0.5584  |   |   |   |   |
> |        | RubikConv | 47.3274 | 0.9885 | 0.0103 | 0.5486  |   |   |   |   |
>
> **2. Comparison with the best algorithms.**
>
> In the manuscript, we conducted experiments on image denoising and de-blurring tasks using Restormer, which achieved performance that closely approximated the best results on both restoration tasks. In addition, we made an effort to integrate the proposed RubikConv into leading algorithms, such as SNR [1] for low-light image enhancement, FFTformer [2] for image de-blurring, and UAPN [3] for guided image resolution. The quantitative results presented below further demonstrate the effectiveness of our method.
>
> | Model | Config    | LOL     |        | Huawei  |         |   |   |   |   |
> |-------|-----------|---------|--------|---------|---------|---|---|---|---|
> |       |           | PSNR    | SSIM   | PSNR    | SSIM    |   |   |   |   |
> |       | Original  | 24.5276 | 0.8407 | 21.4308 | 0.7065  |   |   |   |   |
> | SNR   | Conv1x1   | 24.4360 | 0.8273 | 21.2811 | 0.6906  |   |   |   |   |
> |       | RubikConv | 24.6507 | 0.8426 | 21.5094 | 0.7113  |   |   |   |   |
>
> Table 3: Quantitative comparison of image de-blurring. The model is only trained on the GoPro training set and directly tested on the GoPro testing set, HIDE, and RealBlur datasets.
> | Model     | Config    | GoPro   |        | HIDE    |        | RealBlur-J |         | RealBlur-R  |         |
> |-----------|-----------|---------|--------|---------|--------|------------|---------|-------------|---------|
> |           |           | PSNR    | SSIM   | PSNR    | SSIM   | PSNR       | SSIM    | PSNR        | SSIM    |
> |           | Original  | 34.0694 | 0.9527 | 31.2796 | 0.9476 | 29.5407    | 0.8860  | 36.8165     | 0.9607  |
> | FFTformer | Conv1x1   | 33.6308 | 0.9455 | 30.8762 | 0.9403 | 28.8752    | 0.8789  | 36.1029     | 0.9352  |
> |           | RubikConv | 34.0867 | 0.9533 | 31.2969 | 0.9480 | 29.5549    | .0.8863 | 36.8353     | 0.9612
>
> Table 4: Quantitative comparison of guided image super-resolution.
> | Model | Config    | WroldView-II |        |        |        | GaoFen2  |        |        |         |
> |-------|-----------|--------------|--------|--------|--------|----------|--------|--------|---------|
> |       |           | PSNR         | SSIM   | SAM↓   | ERGAS↓ | PSNR     | SSIM   | SAM↓   | ERGAS↓  |
> |       | Original  | 41.7156      | 0.9657 | 0.0227 | 0.9506 | 47.4635  | 0.9895 | 0.0100 | 0.5382  |
> | UAPN  | Conv1x1   | 41.6826      | 0.9615 | 0.0231 | 0.9557 | 47.4091  | 0.9890 | 0.0104 | 0.5392  |
> |       | RubikConv | 41.7675      | 0.9660 | 0.0220 | 0.9454 | 47.5260  | 0.9898 | 0.0097 | 0.5380  |
>
> [1] Xu X, Wang R, Fu C W, et al. SNR-aware Low-Light Image Enhancement, Proceedings of the IEEE/CVF Conference on Computer Vision and Pattern Recognition, 2022.
>
> [2] Kong L, Dong J, Ge J, et al. Efficient Frequency Domain-based Transformers for High-Quality Image Deblurring, Proceedings of the IEEE/CVF Conference on Computer Vision and Pattern Recognition, 2023.
>
> [3] Zheng K, Huang J, Zhou M, et al. Deep Adaptive Pansharpening via Uncertainty-aware Image Fusion, IEEE Transactions on Geoscience and Remote Sensing, 2023.
>
> **3. Efficiency of the proposed RubikConv.**
>
> We report the model size, FLOPs (an image with 400\*600\*3 pixels), and average running time on the LOL test set (including 15 400\*600\*3 images) in Table 5. The running time is measured on a workstation with an NVIDIA GTX 3090 GPU. We only replace two standard convolution layers in the DRBN baseline with the proposed RubikConv, thus, the extra running time introduced by the RubikConv is negligible. Since the RubikConv only requires convolution with a 1x1 kernel, the FLOPs and parameters are fewer than the baseline while DRBN -RubikConv achieves a 0.58db performance improvement.
>
> Table 5: The quantitative results, FLOPs, and average running time of the DRBN baseline on the LOL test set.
> | Model | Config    | PSNR    | SSIM   | Flops (G) | Running time (s) |   |   |   |   |
> |-------|-----------|---------|--------|-----------|------------------|---|---|---|---|
> |       | Original  | 19.7931 | 0.8361 | 39.037    | 0.256            |   |   |   |   |
> | DRBN  | Conv1x1   | 19.8648 | 0.8340 | 38.445    | 0.255            |   |   |   |   |
> |       | RubikConv | 20.3769 | 0.8400 | 38.563    | 0.263            |   |   |   |   |

---

### Decision · Program_Chairs · 2023-09-21

**Decision:**

Accept (poster)

**Comment:**

This paper is highly evaluated by all reviewers for its good novelty, and sufficient experiments. All review comments have been well addressed in the rebuttal. I thus suggest an acceptance.